# Intention to have the seasonal influenza vaccination during the COVID-19 pandemic among eligible adults in the UK: a cross-sectional survey

Susan M Sherman ⬥,[1] Julius Sim ⬥,[2] Richard Amlôt,[3,4] Megan Cutts,[1] Hannah Dasch,[5,6] G James Rubin ⬥,[3,5] Nick Sevdalis ⬥,[5,6] Louise E Smith ⬥ [3,5]

For numbered affiliations see end of article.

**Correspondence to**
Dr Susan M Sherman;
s.m.sherman@keele.ac.uk

## ABSTRACT

**Objective** To investigate the likelihood of having the seasonal influenza vaccination during the COVID-19 pandemic in individuals who were eligible to receive it.

**Design** We conducted a cross-sectional online survey in July 2020. We included predictors informed by previous research, in the following categories: sociodemographic variables; uptake of influenza vaccine last winter and beliefs about vaccination.

**Participants** 570 participants (mean age: 53.07; 56.3% female, 87.0% white) who were eligible for the free seasonal influenza vaccination in the UK.

**Results** 59.7% of our sample indicated they were likely to have the seasonal influenza vaccination, 22.1% reported being unlikely to have the vaccination and 18.2% were unsure. We used logistic regression to investigate variables associated with intention to receive a seasonal influenza vaccine in the 2020–2021 season. A positive attitude to vaccination in general predicted intention to have the influenza vaccine in 2020–2021 (OR 1.45, 95% CI 1.19 to 1.77, p<0.001) but the strongest predictor of intention was previous influenza vaccination behaviour (OR 278.58, 95% CI 78.04 to 994.46, p<0.001).

**Conclusions** Previous research suggests that increasing uptake of the influenza vaccination may help contain a COVID-19 outbreak, so steps need to be taken to convert intention into behaviour and to reach those individuals who reported being unlikely or unsure about having the vaccine.

## INTRODUCTION

To maximise uptake and help contain subsequent COVID-19 and other infectious disease outbreaks, we need to understand influences on intention to have the influenza vaccination while COVID-19 is circulating. We report findings from a survey conducted in July 2020 in the UK, which explored participants' likelihood of having the seasonal influenza vaccination in 2020–2021.

The COVID-19 pandemic was declared on 11 March 2020. While the first wave of the pandemic missed most of the influenza season in the Northern hemisphere, a second

### Strengths and limitations of this study

► First study to explore seasonal influenza vaccine acceptability in the UK during the COVID-19 pandemic.
► Comprehensive demographical information was collected to facilitate statistical analysis of eligible subgroups.
► Survey measured intention rather than behaviour.

wave has overlapped with the 2020–2021 season.[1] Healthcare systems come under considerable strain during a typical influenza season; this has been compounded by a large number of COVID-19 cases. Recent research has modelled the impact of mass influenza vaccination on the spread of COVID-19, in the event of such an overlap, and suggests that increasing uptake of the influenza vaccination would facilitate efforts to contain COVID-19 outbreaks.[2] In addition, there is some evidence to suggest that patients with a recent history of influenza or influenza-like illnesses are at risk of more severe COVID-19.[3] However, increasing, or even maintaining, levels of influenza vaccination may be problematic if reduced uptake patterns seen already in other vaccines also hold for the influenza vaccine. For example, the uptake of the measles-mumps-rubella (MMR) vaccine in England became 19.8% lower in the 3 weeks after full physical distancing measures were introduced in March than it was for the same period in 2019.[4]

The influenza season in the UK runs from December until March each year and the national vaccination programme starts in September. At the time of data collection, eligibility for the free vaccine through the National Health Service (NHS) was the same as in previous years, being available to children aged 2–11, adults over 65, pregnant women, health and social care workers,



individuals aged 6 months to 65 years who are in clinical at-risk groups (many of which coincide with the COVID-19 at-risk groups), those living in a residential or nursing home and anyone who is the main carer of an older or disabled person. In November 2020, the 2020–2021 influenza vaccination programme was extended to all adults aged 50 or older as well as to anyone living with someone who is at high risk from coronavirus. The vaccination is also available privately for a charge through primary care and pharmacies to the rest of the population. Despite the wide availability of a free vaccine for eligible individuals, uptake varies across the different categories of eligibility; for example, in the 2019–2020 season 72.4% of 65+ adults in England were vaccinated compared with 44.9% of individuals aged 6 months to 65 years in clinical at-risk groups.[5]

In order to protect people ahead of and during the annual influenza season, it might be helpful to understand intention to have a seasonal influenza vaccination during the COVID-19 pandemic. To this end, we explored participants' likelihood of having the seasonal influenza vaccination as part of a larger cross-sectional study investigating attitudes towards a potential COVID-19 vaccination.[6] Previous research exploring factors associated with seasonal influenza vaccination uptake has identified a range of factors that might influence seasonal influenza uptake.[7 8] In this study, we focused on sociodemographic factors such as age, ethnicity and gender, general attitude towards vaccination, fear of needles and past behaviour (whether individuals previously had a seasonal influenza vaccination). We explored these factors in participants who were eligible for the free influenza vaccination under pre-COVID criteria (in place at the time of data collection) and conducted a sensitivity analysis in which we included all those who were eligible under the pandemic-motivated broadened criteria.

## MATERIALS AND METHODS

A nationally representative quota sample of 1500 UK adults (quotas set on age, gender and ethnicity) was recruited through Prolific's online research panel to complete a cross-sectional survey between 14 and 17 July 2020. Participants were included in this study if they were eligible to receive the free influenza vaccine through the NHS at the time of data collection (aged 65 years or over, pregnant, working in health or social care or in a clinical risk group). We did not collect data on whether participants were living in a care home or whether they were a main carer, and so these eligibility criteria are not explicitly represented in our analysis. After providing consent, participants were asked to complete the survey, which included: sociodemographic questions (eg, age, gender, ethnicity, employment status, highest educational or professional qualification); clinical questions (eg, whether they or someone else in their household (if applicable) had a chronic illness that made them clinically vulnerable to serious illness from COVID-19); questions about COVID-19 (eg, whether they were worried about catching coronavirus) and questions about a possible COVID-19 vaccination (eg, whether they thought most people would get a coronavirus vaccination). We also asked participants to what extent they agreed that 'in general, vaccination is a good thing' and to what degree they were 'afraid of needles' (both on an 11-point scale from 'strongly disagree' to 'strongly agree'), and if they had been vaccinated for seasonal influenza last winter (yes/no). The outcome measure for this study, influenza vaccination intention, was measured by asking participants how likely they would be to have the seasonal influenza vaccine 'this winter' (11-point scale, from 'extremely unlikely' to 'extremely likely'). Full details of the wider study, including survey methodology, are reported elsewhere.[6] The survey is available in online supplemental file 1.

Since eligibility for the free influenza vaccine in the 2020–2021 season was widened after data collection, as a sensitivity analysis, we re-ran analyses using these broader criteria.

## Statistical analysis

In order to identify factors associated with intention to receive the seasonal influenza vaccine in the 2020–2021 season, we used a multivariable logistic regression model, based on those respondents expressing a clear intention either to have or not to have the vaccine. The predictors in the model were specified a priori, based on previous research[7 8]: sociodemographic variables; uptake of influenza vaccine last winter and beliefs about vaccination (value of vaccination in general; afraid of needles); see table 1. Odds ratios (ORs) with 95% CIs are reported, adjusted for all of the other predictors in the model; in addition, the corresponding crude (bivariate) ORs are given for the purpose of comparison. We used the Nagelkerke pseudo-$R^2$ statistic to express the goodness-of-fit of the model. Statistical significance was set at $p \leq 0.05$.

## Patient and public involvement

Patients or the public were not involved in the design, or conduct, or reporting or dissemination plans of this research.

## RESULTS

At the time of data collection, 570 individuals in our sample were eligible for the free influenza vaccine. The distribution of influenza vaccination intention was bimodal, with the majority of responses clustering at both ends of the scale. We therefore dichotomised this variable as 0–2 = 'no' (n=126; 22.1%) and 8–10 = 'yes' (n=340; 59.7%) on the 0–10 scale. The 466 respondents who expressed a clear intention either to have or not to have the seasonal influenza vaccine were included in the analysis, and the 104 (18.2%) indeterminate cases were not analysed further. The results of the regression analysis are shown in table 2.

**Table 1** Characteristics and attitudes of those respondents who expressed a clear intention either to have or not to have the influenza vaccine (n=466)

| Variable | |
|---|---|
| Age (years): mean (SD); range | 53.07 (16.86); 18–87 |
| Gender* | |
| Male | 203 (43.7) |
| Female | 262 (56.3) |
| Ethnicity† | |
| White | 403 (87.0) |
| Non-white | 60 (13.0) |
| Qualifications | |
| Degree equivalent or higher | 231 (49.6) |
| Other | 235 (50.4) |
| Working‡ | |
| Part-time | 58 (12.5) |
| Full-time | 136 (29.3) |
| Not working/other§ | 270 (58.2) |
| Key worker | |
| Yes | 175 (37.6) |
| No | 291 (62.4) |
| Influenza vaccination last winter* | |
| Yes | 299 (64.3) |
| No | 166 (35.7) |
| In general, vaccination is a good thing (0–10 scale): mean (SD)¶ | 8.78 (2.09) |
| I am afraid of needles (0–10 scale): mean (SD)¶ | 2.14 (3.17) |

Values are n (%) except where stated otherwise.
*One missing value.
†Three missing values.
‡Two missing values.
§Includes 30 unemployed, 36 furloughed and 157 retired.
¶These variables showed a marked skew.

A positive attitude to vaccination and previous vaccination behaviour were significant predictors of intention to have the influenza vaccine in the 2020–2021 season. As indicated by the large OR, previous influenza vaccination behaviour was a markedly stronger predictor.

### Sensitivity analysis

We conducted the above analyses based on participants who were eligible for the free influenza vaccine at the time of data collection. In the intervening time, these criteria were broadened. Using the broadened eligibility criteria, 1003 respondents were eligible in 2020–2021. Of these, 491 (49.0%) respondents expressed a clear intention to have the vaccine, 291 (29.0%) had a clear intention not to have the vaccine and there were 221 (22.0%) indeterminate cases. As before, the indeterminate cases

were not analysed. The results of the sensitivity analysis are shown in table 3.

Compared with the main analysis, a somewhat smaller percentage of respondents indicated a clear intention to be vaccinated (49.0% vs 59.7%), and correspondingly a larger percentage not to be vaccinated (29.0% vs 22.1%). However, the ORs were of a similar magnitude to those in the main analysis and with similar associated p values (though with the larger sample size, the OR for age became significant), suggesting that while the broader eligibility criteria may influence the percentage of individuals intending to be vaccinated against influenza, they have little effect on the predictors of such vaccination behaviour.

In both the main analysis and the sensitivity analysis, the adjusted and crude ORs were similar, with the exception of those for ethnicity. For this variable, the ORs changed noticeably (and went from being significant to non-significant) after adjustment for the other predictors in the multivariable model, suggesting that some of the explanatory effect of ethnicity was redistributed to other predictors in the full model.

### DISCUSSION

These findings strongly suggest that individuals who had the influenza vaccine in the last influenza season were likely to intend to have it again in the 2020–2021 season. This is consistent with findings from the H1N1 influenza pandemic[7] as well as with findings from studies exploring influenza vaccination intentions during the COVID-19 pandemic in other countries and regions such as Italy[9 10] and Catalonia.[11] It also aligns with the finding that across six countries (USA, Canada, Israel, Japan, Spain, Switzerland) parents' intention to vaccinate their child against seasonal influenza was influenced by their and their child's previous influenza vaccination status.[12] However, there are still key issues to address. Vaccination intention across all those individuals in our sample who were eligible for the vaccine (59.7%) at the time of data collection was slightly lower than the reported uptake from the last influenza season (64.3%). Furthermore, it is likely that actual uptake will be lower than intention as a result of the intention-behaviour gap,[13] making it important that efforts are made to convert positive intentions into uptake. This might be achieved through appropriate messaging and special arrangements for vaccine delivery, particularly for those who might be shielding or at higher risk from COVID-19 and reluctant to attend their general practitioner's surgery. Both approaches are also likely to be needed to motivate those individuals who have not previously had the influenza vaccine and those individuals who are eligible for free vaccination but who were among respondents in our sample who indicated they definitely did not intend to be vaccinated (22.1%) or were unsure (18.2%).

Limitations of the current study include that participants were reporting intention to be vaccinated rather

**Table 2** Logistic regression analysis of variables associated with intention to receive a seasonal influenza vaccine

|  | Adjusted (crude) OR | 95% CI | P value |
|---|---|---|---|
| Age (years) | 1.02 (1.03) | 0.99 to 1.04 | 0.190 |
| Gender (reference: female) | 1.24 (1.37) | 0.60 to 2.59 | 0.565 |
| Ethnicity—white (reference: black and minority ethnic) | 0.60 (1.99) | 0.23 to 1.53 | 0.281 |
| Qualifications—degree equivalent or higher (reference: other) | 1.52 (0.94) | 0.72 to 3.22 | 0.278 |
| Working (reference: not working/other) |  |  |  |
| Part-time | 1.12 (0.75) | 0.39 to 3.21 | 0.830 |
| Full-time | 1.51 (0.81) | 0.64 to 3.56 | 0.342 |
| Key worker (reference: not key worker) | 0.59 (0.62) | 0.27 to 1.31 | 0.197 |
| Influenza vaccination last winter (reference: no vaccination) | 278.58 (273.58) | 78.04 to 994.46 | <0.001 |
| In general, vaccination is a good thing (0–10 scale) | 1.45 (1.57) | 1.19 to 1.77 | <0.001 |
| I am afraid of needles (0–10 scale) | 0.98 (.091) | 0.89 to 1.09 | 0.757 |

Reference categories for the ORs are shown where appropriate. CIs and p values relate to the adjusted ORs. n=460 (six cases with missing data on one or more variables were not analyzed). Nagelkerke $R^2$=0.760.

**Table 3** Logistic regression analysis of variables associated with intention to receive a seasonal influenza vaccine, using the broadened eligibility criteria introduced in the 2020–2021 season

|  | Adjusted (crude) OR | 95% CI | P value |
|---|---|---|---|
| Age (years) | 1.02 (1.02) | 1.00 to 1.04 | 0.046 |
| Gender (reference: female) | 1.24 (1.12) | 0.76 to 2.01 | 0.388 |
| Ethnicity—white (reference: black and minority ethnic) | 0.75 (1.73) | 0.36 to 1.56 | 0.443 |
| Qualifications—degree equivalent or higher (reference: other) | 1.00 (1.02) | 0.62 to 1.59 | 0.983 |
| Working (reference: not working/other) |  |  |  |
| Part-time | 0.80 (0.68) | 0.40 to 1.58 | 0.519 |
| Full-time | 1.03 (0.71) | 0.58 to 1.81 | 0.929 |
| Key worker (reference: not key worker) | 0.93 (0.83) | 0.55 to 1.57 | 0.776 |
| Influenza vaccination last winter (reference: no vaccination) | 281.78 (262.85) | 95.35 to 832.72 | <0.001 |
| In general, vaccination is a good thing (0–10 scale) | 1.54 (1.57) | 1.31 to 1.81 | <0.001 |
| I am afraid of needles (0–10 scale) | 0.98 (0.92) | 0.91 to 1.05 | 0.529 |

Reference categories for the ORs are shown where appropriate. CIs and p values relate to the adjusted ORs. n=774 (eight cases with missing data on one or more variables were not analyzed). Nagelkerke $R^2$=0.716.

than actual vaccination status and that they were collected before a COVID-19 vaccination was a reality. However, since public health systems globally are likely to be managing seasonal influenza against a backdrop of COVID-19 for the foreseeable future, we believe these data provide useful information to assist with understanding the evolving response to national vaccination programmes.

The NHS is often overwhelmed during the influenza season, needing, for example, to cancel routine operations. The extension of the influenza vaccination programme in 2020–2021 to people aged 50 years and over and to those living with someone who is at high risk from coronavirus may also help decrease the burden of the influenza season. Potential carry-over effects into the next influenza season (2021–2022 in the UK), in light of the current availability of COVID-19 vaccination, and

ongoing public health and media discussions regarding the need for seasonal vaccination programmes for both corona and influenza viruses suggest that the current dataset can be used as a baseline for future evaluation of the uptake of the influenza vaccination within an ever-changing context. Increasing uptake of the seasonal influenza vaccine in a timely fashion will relieve pressure on health services. If this is to be successful, strategies to achieve this increase need to be designed now.

**Author affiliations**
[1]School of Psychology, Keele University, Keele, UK
[2]School of Medicine, Keele University, Keele, UK
[3]NIHR Health Protection Research Unit in Emergency Preparedness and Response, King's College London, London, UK
[4]Emergency Response Department Science and Technology, Public Health England, Salisbury, UK

⁵Institute of Psychiatry, Psychology and Neuroscience, King's College London, London, UK
⁶Centre for Implementation Science, King's College London, London, UK

**Contributors** Conceptualisation, LES; Methodology, RA, MC, HD, GJR, NS, SMS, JS, LES; Software, SMS; Formal analysis, JS; Writing—original draft preparation, SMS; Writing—review and editing, RA, MC, HD, GJR, NS, SMS, JS, LES; Funding acquisition, RA, GJR, NS, SMS, JS, LES.

**Funding** Data collection was funded by a Keele University Faculty of Natural Sciences Research Development award to SMS, JS and NS, and a King's Together Rapid COVID-19 award granted jointly to LES, GJR, RA, NS, SMS and JS. LES, RA and GJR are supported by the National Institute for Health Research Health Protection Research Unit (NIHR HPRU) in Emergency Preparedness and Response, a partnership between Public Health England, King's College London and the University of East Anglia. NS's research is supported by the National Institute for Health Research (NIHR) Applied Research Collaboration (ARC) South London at King's College Hospital NHS Foundation Trust. NS is a member of King's Improvement Science, which offers co-funding to the NIHR ARC South London and comprises a specialist team of improvement scientists and senior researchers based at King's College London. Its work is funded by King's Health Partners (Guy's and St Thomas' NHS Foundation Trust, King's College Hospital NHS Foundation Trust, King's College London and South London and Maudsley NHS Foundation Trust), Guy's and St Thomas' Charity and the Maudsley Charity.

**Disclaimer** The views expressed are those of the authors and not necessarily those of the NIHR, the charities, Public Health England or the Department of Health and Social Care.

**Competing interests** NS is the director of the London Safety and Training Solutions Ltd, which offers training in patient safety, implementation solutions and human factors to healthcare organisations and the pharmaceutical industry.

**Patient consent for publication** Not required.

**Ethics approval** Ethical approval for this study was granted by Keele University's Research Ethics Committee (reference: PS-200129).

**Provenance and peer review** Not commissioned; externally peer reviewed.

**Data availability statement** Data are available in a public, open access repository. Survey items and dataset are available from: DOI 10.17605/OSF.IO/94856.

**ORCID iDs**
Susan M Sherman http://orcid.org/0000-0001-6708-3398
Julius Sim http://orcid.org/0000-0002-1816-1676
G James Rubin http://orcid.org/0000-0002-4440-0570
Nick Sevdalis http://orcid.org/0000-0001-7560-8924
Louise E Smith http://orcid.org/0000-0002-1277-2564

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
