## [Reviewer comments · BMJ Open]

ARTICLE DETAILS

TITLE (PROVISIONAL)	Intention to have the seasonal influenza vaccination during the COVID-19 pandemic among eligible adults in the UK: A cross-sectional survey
AUTHORS	Sherman, Susan; Sim, Julius; Amlot, Richard; Cutts, Megan; Dasch, Hannah; Rubin, GJ; Sevdalis, Nick; Smith, Louise

VERSION 1 – REVIEW

REVIEWER	Gabriella d'Ettoire University of Rome Sapienza, Public health and infectious diseases
REVIEW RETURNED	01-Feb-2021

GENERAL COMMENTS	Thanks for the opportunity to review the manuscript titled "Intention to have the seasonal influenza vaccination during the COVID-19 pandemic among eligible adults in the UK: A cross-sectional survey". The research addresses an important and relevant issue regarding the seasonal influenza vaccination during the COVID-19 pandemic and explored participants' likelihood of having such vaccination. The results are presented in a comprehensive way and provide new insights for future interventions aimed to decrease the burden of seasonal influenza during COVID-19 outbreak.. The layout and the format of the manuscript are excellent. Policy-makers and healthcare workers could benefit from reading this manuscript. The authors may point out that recent findings showed that patients with history of recent influenza or influenza like illnesses has been found to incur greater risk and severity of Covid-19; therefore previous influenza or ILI could represent a predisposing factor for subsequent Covid-19 infection (G. Ceccarelli, G. d'Ettoire, C.M. Mastroianni, G. d'Ettoire, "Is previous influenza-like illness a potential trojan horse for COVID-19?" Crit. Care (2020), https://doi.org/10.1186/s13054-020-03226-5. Given these findings, vaccination against influenza should be provided both to minimize the individual susceptibility to Covid-19 and to avoid superimposed infection from influenza viruses in patients suffering from Covid-19. Limitations of the study should be stated.
---

REVIEWER	Angela Dominguez University of Barcelona, Dept of Public Health
REVIEW RETURNED	10-Feb-2021

GENERAL COMMENTS	General comment: The manuscript "Intention to have the seasonal influenza vaccination during the COVID-19 pandemic among eligible adults in the UK: A cross-sectional survey" has been reviewed. The topic of this study is interesting because knowledge about attitude towards vaccination in general and to influenza vaccination in particular are needed to optimize the use of vaccines. However,
--

	there are several points that authors should improve (see specific comments). Specific comments: Page 3, line 21: "2020/21" should be changed to "the 2020-2021 season" Page 3, lines 26 to 28: the point estimate of the Odds ratio should be included (not only the 95% confident limits) Page 4, line 7: "this year" should be deleted Page 4, line 25: "a COVID-19 outbreak" should be changed to "COVID-19 outbreaks" Page 4, line 57: "it is helpful" might be changed to "it might be helpful" Page 5, lines 7 to 12: In fact in this study, authors are comparing the factors associated to receive the influenza vaccine in previous seasons with the factors associated to receive the influenza vaccine using the broadened eligibility criteria introduced in the 2020-2021 season. This should be included in the aim of the study. Page 5, line 25: "pregnant" should be changed to "pregnancy" Page 5: The statistical analysis used to assess the results obtained should be explained, with special mention on how the crude odds ratio and the adjusted odds ratio were estimated. The level of statistical significance considered should be included in this section. Page 6, line 20: "in 2020-2021" should be changed to "in the 2020-2021 season" Page 8, lines 5 and 11: "in 2020-2021" should be changed to "in the 2020-2021 season" Page 9, line 11: "this year" should be changed to "in the 2020-2021 season" Page 9, line 16: "from last year" should be changed to "from last influenza season" Discussion: Because the factors associated to receive the influenza vaccine in previous seasons were compared with the factors associated to receive the influenza vaccine using the broadened eligibility criteria introduced in the 2020-2021 season, authors should compare the results obtained with those of other authors investigating the influence of an extension of the free vaccination program to other groups of people. In fact, in the current version only two studies (references 6 and 8) are mentioned in the discussion section. Limitations of the study should be mentioned and how they might influence the results obtained Tables: In table 1 the title is "Characteristics and attitudes of those respondents who expressed a clear intention either to have or not have the influenza vaccine, n= 466", but in page 7 it is stated "In our sample, 1003 respondents met the broadened eligibility criteria for the free influenza vaccine in 2020-2021. Of these, 491 respondents expressed a clear intention to have the vaccine, 291 a clear intention not to have the vaccine, and there were 221 indeterminate cases. Therefore, those who expressed a clear intention to have or not to have the influenza vaccine were 491 + 291 = 782. In table 2 and table 3 crude and adjusted odds ratio should be included; the sentence "statistical significance was set at $p \leq 0.05$" should be taken out of the title and included in the statistical analysis section.
--	---

REVIEWER	Michael Stoto Georgetown University
REVIEW RETURNED	14-Feb-2021

GENERAL COMMENTS	This is a very well-done piece of survey research on a critical global issue. Unfortunately, however, the results have been overtaken by events. The U.K, is now well into it's COVID-19 vaccine campaign, and apparently doing well compared to other countries (at least in terms of total numbers). People's attitudes in July last year are no longer particularly relevant. Although I realize that this is a fundamentally different piece of research, it would be interesting to see how attitudes changed when vaccines did become available, or at least how the factors that predicted vaccine uptake in the survey ended up explaining it (or not) when the vaccine became available.
---

VERSION 1 – AUTHOR RESPONSE

Reviewer: 1 Dr. Gabriella d'Ettorre, University of Rome Sapienza

Thanks for the opportunity to review the manuscript titled "Intention to have the seasonal influenza vaccination during the COVID-19 pandemic among eligible adults in the UK: A cross-sectional survey". The research addresses an important and relevant issue regarding the seasonal influenza vaccination during the COVID-19 pandemic and explored participants' likelihood of having such vaccination. The results are presented in a comprehensive way and provide new insights for future interventions aimed to decrease the burden of seasonal influenza during COVID-19 outbreak.. The layout and the format of the manuscript are excellent. Policy-makers and healthcare workers could benefit from reading this manuscript.

Thank you for your positive appraisal of our work – it is much appreciated.

The authors may point out that recent findings showed that patients with history of recent influenza or influenza like illnesses has been found to incur greater risk and severity of Covid-19; therefore previous influenza or ILI could represent a predisposing factor for subsequent Covid-19 infection (G. Ceccarelli, G. d'Ettorre, C.M. Mastroianni, G. d'Ettorre, "Is previous influenza-like illness a potential trojan horse for COVID-19?" Crit. Care (2020), <https://doi.org/10.1186/s13054-020-03226-5>. Given these findings, vaccination against influenza should be provided both to minimize the individual susceptibility to Covid-19 and to avoid superimposed infection from influenza viruses in patients suffering from Covid-19.

This has now been added.

Limitations of the study should be stated.

These have now been added.

Reviewer: 2 Dr. Angela Dominguez, University of Barcelona

General comment: The manuscript "Intention to have the seasonal influenza vaccination during the COVID-19 pandemic among eligible adults in the UK: A cross-sectional survey" has been reviewed. The topic of this study is interesting because knowledge about attitude towards vaccination in general and to influenza vaccination in particular are needed to optimize the use of vaccines.

We thank you for your positive comments about our work.

However, there are several points that authors should improve (see specific comments).

Specific comments:

Page 3, line 21: “2020/21” should be changed to “the 2020-2021 season”

Changed

Page 3, lines 26 to 28: the point estimate of the Odds ratio should be included (not only the 95% confident limits)

This has now been added.

Page 4, line 7: “this year” should be deleted

Changed

Page 4, line 25: “a COVID-19 outbreak” should be changed to “COVID-19 outbreaks”

Changed

Page 4, line 57: “it is helpful” might be changed to “it might be helpful”

Changed

Page 5, lines 7 to 12: In fact in this study, authors are comparing the factors associated to receive the influenza vaccine in previous seasons with the factors associated to receive the influenza vaccine using the broadened eligibility criteria introduced in the 2020-2021 season. This should be included in the aim of the study.

To clarify: we do not explore the factors associated with previous influenza vaccination, but we have added the broadened criteria element into the aims as you recommended.

Page 5, line 25: “pregnant” should be changed to “pregnancy”

Since this refers to their status of being pregnant (i.e. an adjective) it would be ungrammatical to change this, so we propose to keep the word as it stands.

Page 5: The statistical analysis used to assess the results obtained should be explained, with special mention on how the crude odds ratio and the adjusted odds ratio were estimated. The level of statistical significance considered should be included in this section.

A separate section on statistical analysis has been included, including information on the estimation of the odds ratios.

Page 6, line 20: “in 2020-2021” should be changed to “in the 2020-2021 season”

Changed

Page 8, lines 5 and 11: “in 2020-2021” should be changed to “in the 2020-2021 season”

Changed

Page 9, line 11: “this year” should be changed to “in the 2020-2021 season”

Changed

Page 9, line 16: “from last year” should be changed to “from last influenza season”

Changed

Discussion:

Because the factors associated to receive the influenza vaccine in previous seasons were compared with the factors associated to receive the influenza vaccine using the broadened eligibility criteria introduced in the 2020-2021 season, authors should compare the results obtained with those of other authors investigating the influence of an extension of the free vaccination program to other groups of people. In fact, in the current version only two studies (references 6 and 8) are mentioned in the discussion section.

We have expanded the Discussion to contextualise the findings to other studies.

Limitations of the study should be mentioned and how they might influence the results obtained

These have now been added.

Tables:

In table 1 the title is “Characteristics and attitudes of those respondents who expressed a clear intention either to have or not have the influenza vaccine, n= 466”, but in page 7 it is stated “In our sample, 1003 respondents met the broadened eligibility criteria for the free influenza vaccine in 2020-2021. Of these, 491 respondents expressed a clear intention to have the vaccine, 291 a clear intention not to have the vaccine, and there were 221 indeterminate cases. Therefore, those who expressed a clear intention to have or not to have the influenza vaccine were 491 + 291 = 782.

To clarify: table 1 reports findings from those participants who were identified using the original criteria and not the broadened criteria, which was used for the analysis on page 7. We have added in wording to make this clearer.

In table 2 and table 3 crude and adjusted odds ratio should be included; the sentence “statistical significance was set at $p \leq 0.05$ ” should be taken out of the title and included in the statistical analysis section.

This has been done. The crude ORs have been added, but only the p values and confidence intervals relating to the adjusted ORs are shown.

Reviewer: 3 Prof. Michael Stoto, Georgetown University

This is a very well-done piece of survey research on a critical global issue. Unfortunately, however, the results have been overtaken by events. The U.K. is now well into its COVID-19 vaccine campaign, and apparently doing well compared to other countries (at least in terms of total numbers). People's attitudes in July last year are no longer particularly relevant.

Thank you for your positive comment about our work. While we take your point that events have moved on, we believe that based on current knowledge of COVID, it is likely that the virus will become endemic, circulating seasonally in years to come. Therefore, public health systems globally are likely to take steps to reduce strain on their healthcare services over the winter months, with ongoing seasonal influenza vaccination thus remaining a key priority. As such we feel that these data provide a useful baseline against which future datasets can be evaluated. We believe that the insights gained from our analyses are likely to have continued relevance for any seasonal vaccination program – we have commented on this further in the discussion.

Although I realize that this is a fundamentally different piece of research, it would be interesting to see how attitudes changed when vaccines did become available, or at least how the factors that predicted vaccine uptake in the survey ended up explaining it (or not) when the vaccine became available.

Thank you for this observation, which we fully share. We are in the process of collecting further data longitudinally to allow us to carry out and report such comparative analyses. We have indicated this in the discussion.

VERSION 2 – REVIEW

REVIEWER	Gabriella d'Ettorre University of Rome Sapienza, Public health and infectious diseases
REVIEW RETURNED	10-Mar-2021

GENERAL COMMENTS	The Author fully answered my questions.
---

REVIEWER	Michael Stoto Georgetown University
REVIEW RETURNED	07-Mar-2021

GENERAL COMMENTS	Having read the authors' response to my comments on the original version of the paper, my original opinion remains. I'm not surprised to see that the other reviewers found the paper to be methodologically strong, as did I. And if the authors do, as they suggest, conduct longitudinal data, the July 2020 data will indeed provide a useful benchmark. However, I still do not believe that the results of a survey conducted last year, by themselves, should be published in a scientific journal. Perhaps this is more of a call for the editors than the peer reviewers, but my recommendation is to not publish the paper.
---